



# Quantifying particulate matter optical properties and flow rate in industrial stack plumes from PRISMA hyperspectral imager

Gabriel Calassou[1], Pierre-Yves Foucher[1], and Jean-François Léon[2]

[1]ONERA "The French Aerospace Lab", Département Optique et Techniques Associées (DOTA), 2 avenue Edouard Belin, 31055 Toulouse, France

[2]Laboratoire d'Aérologie, CNRS Université Toulouse 3, 14 avenue Edouard Belin, 31400 Toulouse, France

**Correspondence:** Jean-François Léon (jean-francois.leon@aero.obs-mip.fr)

**Abstract.** Industrial activities such as metallurgy, coal and oil combustion, cement production and petrochemistry release aerosol particles into the atmosphere. We propose to analyze the aerosol composition of plumes emitted by different industrial stacks using PRISMA (PRecursore IperSpettrale della Missione Applicativa) satellite hyperspectral observations. Three industrial sites have been observed: a coal-fired power plant in Matla, South Africa (imaged on September 25, 2021), a steel plant in Wuhan, China (March 24, 2021), and gas flaring at an oil extraction site in Hassi Messaoud, Algeria (July 9, 2021). Below-plume surface reflectances are constrained using a combination of PRISMA and Sentinel-2/MSI images. The radiative transfer simulations are performed for each scene including the surface, background atmosphere, and plume optical properties. The plume aerosol optical thickness (AOT), particle radius, volume of coarse mode aerosol and soot are then retrieved within the plumes following an optimal estimation framework. The mean plume retrieved AOT at $500\,\mathrm{nm}$ ranges between 0.27 and 1.27 and the median radius between $0.10\,\mathrm{\mu m}$ and $0.12\,\mathrm{\mu m}$. We found a volume fraction of soot of 3.6% and 10.4% in the sinter plant and coal-fired plant plumes, respectively. The mass flow rate of particulate matter at point source estimated by an integrated mass enhancement method varies from $511\,\mathrm{g\,s^{-1}}$ for the flaring emission to $1401\,\mathrm{g\,s^{-1}}$ at the coal-fired plant.

## 1 Introduction

Industrial activities such as metallurgy, coal and oil combustion, cement production and petrochemistry release aerosol particles into the atmosphere. The size and the chemical composition of the particles vary according to the the combustion or industrial processes. Fine particles emitted by coal-fired plant (Huang et al., 2017; Saarnio et al., 2014; Linnik et al., 2019; Zhang et al., 2004) or steel factories (Oravisjärvi et al., 2003; Mbengue et al., 2017; Dall'Osto et al., 2008; Weitkamp et al., 2005; Tsai et al., 2007) are generally enriched in heavy metals. Fine particles are also composed of inorganic matter such as sulfate, nitrate and chloride (Riffault et al., 2015; Brock et al., 2003) and PAH or various organics emitted during incomplete combustion (Leoni et al., 2016). The aforementioned toxic elements released by industries or power plants cause adverse heath effects (Pope and Dockery, 2006; Pope et al., 2015; Brook et al., 2010; Bagate et al., 2006) and damages to the environment (Minkina et al., 2020). Regulations on industrial emissions has led to a reduction in emissions (eg. Directive 2010/75/EU). However, emissions standards and the degree to which they are enforced varies geographically among both high- and low-to-medium income





countries. Although monitoring networks dedicated to the survey of particulate matter atmospheric concentrations exist, the
geographical coverage of such networks also varies geographically.

The operational monitoring of aerosol emissions by stationnary industrial point sources can benefit from satellite imagery.
Heavy industries often use stacks to emit and disperse hot air, particulate matter and gaseous pollutants into the atmosphere
that form visible plumes. Stack plumes can be observed from space using dedicated VIS-SWIR camera however the retrieval
of their aerosol content and properties remains a challenge. Retrieval algorithms (Calassou et al., 2021; Philippets et al.,
2018; Foucher et al., 2019) have been developed for the characterization of industrial stack plume using airborne VIS-SWIR
hyperspectral imagery. The method proposed by Calassou et al. (2021) relies on an optimal estimation method (Rodgers, 2000)
for estimating the plume aerosol optical depth and aerosol modal radius. The algorithm introduces the use of Sentinel-2/MSI
VIS-SWIR images in order to evaluate the surface reflectance. We propose in this paper to apply this methodology to PRISMA
(PRecursore IperSpettrale della Missione Applicativa) hyperspectral acquisitions over selected industrial emitors around the
world.

The selected industrial sites are presented in the following section along with a literature review of the aerosol properties
that could be expected in their stack plumes. We analyse the ability of the proposed methodology to detect the aerosol plume
associated to stack emission and to retrieve the aerosol optical properties within the plume. The estimation of the particulate
mass flow rate emitted by the stacks is also analyzed.

## 2  Selection of point source industrial sites

### 2.1  Coal-fired power plant

Coal-fired plants emit a mixture of different size of aerosol particles. Particles emitted from coal combustion are formed by
primary emission (without phase change) and though nucleation, condensation and coagulation of vaporized species (Ninomiya
et al., 2004; Saarnio et al., 2014). Ash formed during pulverized coal combustion has a bi-modal size distribution (Wu et al.,
1999) resulting from different formation processes and influenced by char composition (Baxter, 1992; Kleinhans et al., 2018).
Several clean-up techniques (e.g. electrostatic precipitator or wet flue gas desulfurization) are implemented at facility-level
in order to mitigate toxic and particulate matter emissions (Bhanarkar et al., 2008). The impletation of mitigation techniques
differs upon national regulations (Xu et al., 2016). Early airborne measurements of the aerosol size distribution within the
plume of a coal-fired plants (Cantrell and Whitby, 1978; Richards et al., 1981) have shown nuclei less than $0.03\,\mu m$, an
accumulation mode between $0.1\,\mu m$ and $1.0\,\mu m$, and a coarse mode between $2.0\,\mu m$ and $7.0\,\mu m$ mainly composed of fly ash.
Ehrlich et al. (2007) have found a coarse mode between $6\,\mu m$ and $7\,\mu m$ while recent observations in smokestack plumes of
coal-fired plants in South Korea (Shin et al., 2022) indicate a coarse mode of particles having a diameter between $2.25\,\mu m$ and
$4.50\,\mu m$. Shin et al. (2022); Ehrlich et al. (2007) have observed an accumulation mode with an aerodynamic diameter of $0.6\,\mu m$
and geometrical standard deviation of 1.3. The mass fraction of the fine mode aerosol is between 48% and 62% (Saarnio et al.,
2014; Ehrlich et al., 2007). The aerosol fine fraction emitted by coal-fired plant is mainly composed of water-soluble species
like $SO_4^{2-}$ (21 %) and Ca (26 %) (Saarnio et al., 2014).





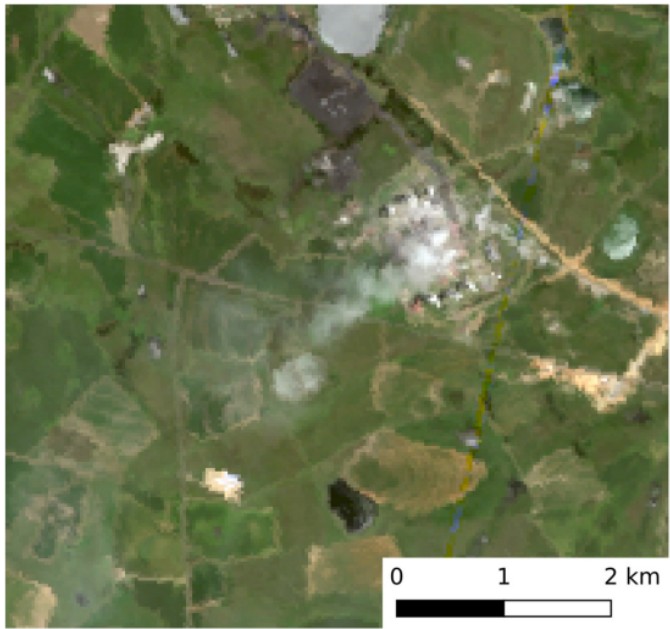

**Figure 1.** PRISMA acquisition over Matla power station on 12th February 2022.

The power station of Matla (S26° 16′ 52.8″,E29° 8′ 29.4″) is located at Kriel, Mpumalanga, South Africa. The power station own by ESKOM has an installed capacity of $6\times600\,\mathrm{MW}$ units. The planned retirement is between 2030 and 2034. The PRISMA image acquired on 12th February 2022 over Matla (Figure 1) shows a distinct aerosol plume advected over a vegetated
area following a N-E axis.

## 2.2  Sinter plant

An iron and steel producing site is a complex of related plants that emit both stack and fugitive particulate matter. The sintering process is a major source of particulate matter and heavy metals. A sinter plant emits particles that are greater in quantity and finer in particle size than other steelmaking emissions (Abreu et al., 2015). Particles are composed of $SO_2^{-4}$, $NO_3$, Na, K, Mg
and $Ca^{2+}$, $NH_4^+$ and trace metals (Almeida et al., 2015; Sylvestre et al., 2017). The chemical profile of particles emitted by a sinter stack is enriched in $K^+$, $Cl^-$, Na and Pb (Hleis et al., 2013). Black carbon (BC) and organic carbon are also detected in the sinter stack emissions (Tsai et al., 2007; Guinot et al., 2016). Following Leoni et al. (2016) the aerosol size distribution is composed of three modes having aerodynamic diameters of $0.1\,\mu m$, $0.6\,\mu m$ and $6\,\mu m$, respectively. The fine mode fraction can be estimed between 30 % et 65 % based on the PM fractions measured by Ehrlich et al. (2007) and Almeida et al. (2015).
China produces about half of the world's steel (Bo et al., 2021). The emission of air pollutant by the steel industry in China has been responsible for air quality degradation and human health problems leading to the introduction of strengthened emission standards (Tang et al., 2020). Wuhan Iron and Steel Co., Ltd. (N30° 38′ 24″, E114° 27′ 36″) is an integrated steel plant



in Wuhan, Hubei province in China. Its nominal crude steel capacity is 15,910 t/year. The factory is located in a peri-urban zone composed of residential areas and agricultural plots.

The plume observed on PRISMA image on 24th march of 2021 (Figure 2) is located at the East of the steelmaking industry and move northeastward. The landscape is heterogeneous, being a mix of residential housing and fields.

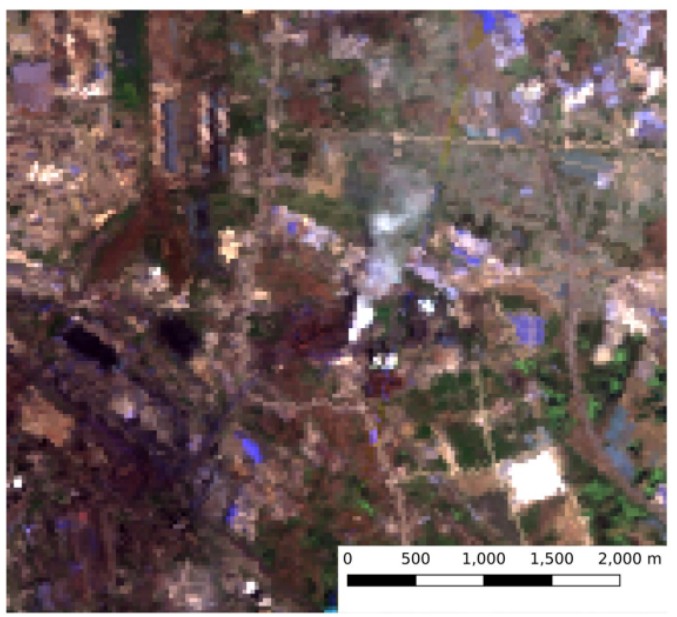

**Figure 2.** PRISMA acquisition over Wuhan (China) on 24th March 2021.

## 2.3   Oil flaring

Flaring is the act of burning off excess gas from oil wells. Flaring happens for economical reasons when excess gas can't be stored and sent elsewhere or for security reason. Flaring from oil wells (also called upstream flaring or flaring of associated
gas) is a significant source of greenhouse gases, aerosol particles and precursors of particles (Klimont et al., 2017).

The incomplete combustion of the flared gas produces soot consisting of mass-fractal-like aggregates of BC containing nanoscale spherules (Fung, 1990). The associated emission factor for BC ranges between $0.13\,\mathrm{g\,m^{-3}}$ (Schwarz et al., 2015; Weyant et al., 2016) to $1.6\,\mathrm{g\,m^{-3}}$ gas flared (Stohl et al., 2013). Flared gas volumes in terms of methane equivalent are globally estimated using the method proposed by Elvidge et al. (2016) and based on VIIRS flare detection (Elvidge et al.,
2013). However the BC emission factor can vary significantly between different oil and gas field (Huang and Fu, 2016). BC strongly absorbs visible light (Bond and Bergstrom, 2006; Bond et al., 2013). Its density is estimated between $1.7\,\mathrm{g\,m^{-3}}$ (Kondo et al., 2011) and $1.9\,\mathrm{g\,cm^{-3}}$ (Medalia and Richards, 1972; Janzen, 1980). The diameter of soot particle emitted by flaring ranges between $10\,\mathrm{nm}$ to $200\,\mathrm{nm}$ and most commonly lies between $10\,\mathrm{nm}$ to $50\,\mathrm{nm}$ (Fawole et al., 2016).





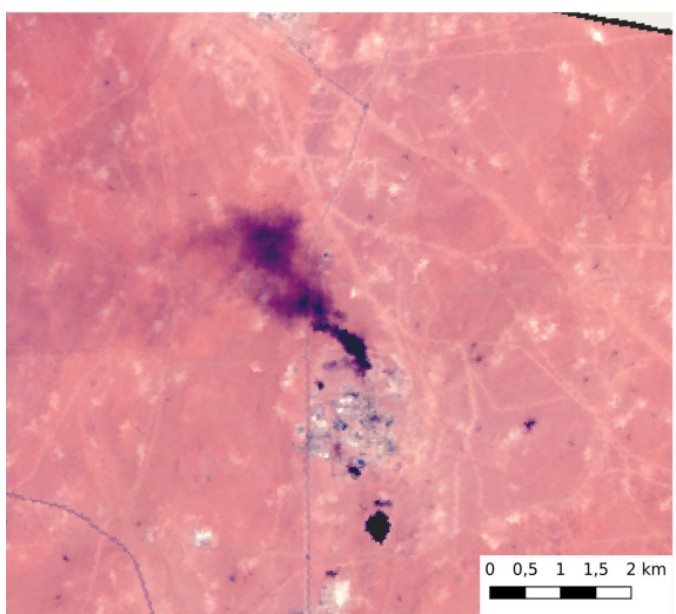

**Figure 3.** PRISMA acquisition over Hassi Messaoud (Algeria) on 9th July 2021.

Hassi Messaoud oil field (N31° 80′, E6° 5′) is located in Ouargla province in Algeria. It's production is centered on $56000\,\mathrm{m^3\,d^{-1}}$.
Hassi Messaoud is a hotspot of gas flaring in the world (Lauvaux et al., 2022; Guanter et al., 2021). A PRISMA image was
acquired on 9th July 2021 at 11:00 LT showing 3 distinct plumes, 2 localed south of the main city and one in the north. The
plumes are easily detectable due to the high contrast between the dark smoke and the bright reflective desertic landscape. The
northern plume (Figure 3) has a northwest orientation, a length of $3\,\mathrm{km}$ and an area of $2\,\mathrm{km^2}$. This plume is emitted by 4 flares.

## 3 Retrieval process

### 3.1 Satellite data and pre-processing

PRISMA mission was launched in March 2019 by the Italian Space Agency for a duration of 5 years. The standard size of
a single image is 30 × 30 km with a ground sampling distance of 30 m. PRISMA is flying a Sun-synchroneous low Earth
orbit at an altitude of 615 km with a local time of equator crossing on descending node at 10:30 (Cogliati et al., 2021). The
hyperspectral imaging spectrometer onboard PRISMA covers the nominal 400-2500 nm spectral range with a sampling interval
between 11 to 15 nm. PRISMA hyspectral data have been used to detect methane plumes linked to oil extraction(Guanter et al.,
2021) and carbon dioxyde emission by power plants (Cusworth et al., 2021).

The aim of the pre-processing is to estimate the surface reflectance below the aerosol plume by a combination of multispectral
SENTINEL-2(S2)/MSI and hyperspectral PRISMA observations. S2/MSI observations are acquired within a few days delay
from PRISMA acquisitions. S2/MSI images were acquired on 27 June 2021, 23 March 2021 and 20 February at 11 LT for the



flaring, sinter plant and coal-fired plant study cases, respectively. S2/MSI has 13 spectral bands between $0.4\,\mu m$ and $2.2\,\mu m$. The pixel resolution is between 10 and 60 m depending on the channel. Both PRISMA and S2/MSI images are first corrected from background atmospheric effects using COCHISE software (Poutier et al., 2002), a front-end of MODTRAN software. The atmospheric parameters used for the atmospheric correction are the ones provided by ESA along with the S2/MSI images (Main-Knorn et al., 2017). The co-registration between both images is operated by using an optical flow algorithm named

GEFOLKI (Brigot et al., 2016). The hyperspectral reflectances below the plume in the PRISMA image are infered from the out-of-plume PRISMA reflectances and the S2/MSI images using the Coupled Non-negative Matrix Factorization (CNMF) technique (Yokoya et al., 2012). The CNMF estimates the hyperspectral surface reflectances below the plume as a linear combination of hyperspectral endmembers weighted by the S2/MSI reflectances. The hyperspectral endmembers are extracted from the hyperspectral image by a vertex composant analysis (VCA) unmixing method (Nascimento and Dias, 2005).

## 3.2 Direct model

The signal measured by a satellite sensor is usually expressed in radiance ($W\,sr^{-1}\,\mu m^{-1}\,m^2$).For a flat, homogeneous and Lambertian ground in a configuration where the environmental effects are neglected, the measured signal can be decomposed as:

$$L_{tot} = L_{atm} + \rho\frac{(E_d + E_s)(T_d + T_s)}{\pi(1 - \rho S)} \tag{1}$$

where $L_{atm}$ is the atmospheric radiance, i.e. the radiance without interaction with the ground, $E_d$ and $E_s$ are respectively the direct and scattering part of the solar irradiance, $T_d$ and $T_s$ are the direct and scattering parts of the atmospheric transmittance, $\rho$ is the surface reflectance and $S$ is the single scattering albedo. For convenience, we normalize the solar illumination by converting the Top of Atmosphere (TOA) radiance $L_{tot}$ into a TOA reflectance signal $\rho_{TOA}$. The conversion from radiance to TOA reflectance is performed by the following equation:

$$\rho_{TOA} = \frac{\pi L_{tot}}{E_{TOA}\mu_s} \tag{2}$$

where $E_{TOA}$ represents the TOA solar irradiance and $\mu_s$ is the cosine of the azimuth solar angle. In the presence of a plume, all the radiative terms except the surface reflectance $\rho$ are affected.

The direct model associated with pixel i for a spectro-imager is based on the COMANCHE software(under plane and parallel atmospheric hypothesis) for the calculation of the different terms $L_{atm}$, $E_d$, $E_s$, $T_d$, $T_s$ and $S$. A first calculation is

done corresponding to the atmospheric conditions of the acquisition, then a second calculation is necessary to introduce the aerosol plume radiative impact. To avoid a complex 3D atmospheric model while taking into account the spatial heterogeneity of the plume concentration, we introduce a dual AOT model. The dual AOT model for a Nadir viewing angle is illustrated in Figure 4. Besides the AOT associated with pixel i noted $\delta$ is associated to the different optical paths in the column above the pixel i (blue lines in Figure 4), a second AOT noted $\delta^*$ is introduced. $\delta^*$ is associated to the optical path of the direct solar flux





intercepting pixel $i$ surface at ground level (red line in Figure 4). Thus $\delta^*$ doesn't correspond to pixel $i$ atmospheric column as illustrated in Figure 4. Depending on the geometry of the plume (height, thickness and direction) and the solar position, $\delta^*$ can take very heterogeneous values: low values for pixels at the edge of the plume when the downward flux doesn't intercept the plume, values close to $\delta$ (pixels at the heart of a fairly extensive plume) or high values at the edge of the plume when the downward flux crosses the plume at its heart. The final direct model for pixel of coordinates $(x_i, y_i)$ with corresponding AOT $\delta$

is:

$$L_{tot} = L_{atm}^\delta + \rho \frac{(E_d^{\delta^*} + E_s^\delta)(T_d^\delta + T_s^\delta)}{\pi(1 - \rho S)} \tag{3}$$

Where, $X^\delta$ means a calculation of $X$ including a plume homogeneous plane parallele layer associated to the AOT value $\delta$.

Our assumption is that these two parameters can be estimated independently using only the radiance observed at pixel $i$ at sensor level: $\delta$ is associated with the extinction and scattering (reflection) properties of the plume while $\delta^*$ is only associated

with the extinction properties of the aerosol plume.

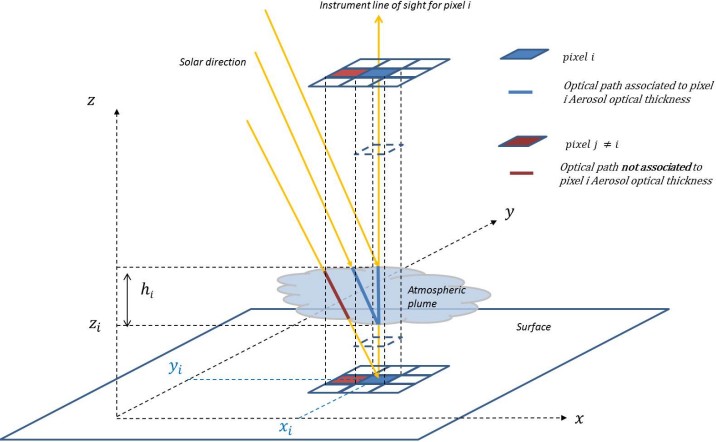

**Figure 4.** Schematic representation of the interaction of solar radiation with a particle plume emitted by a stack.

### 3.3   Optimal estimation formalism

Optimal estimation is a regularized matrix inverse method based on Bayes' theorem (Rodgers, 2000). The measured spectral radiance vector $y$ is modelled as a vector-valued function:

$$y = F(x) + \epsilon \tag{4}$$

where $F$ is the forward model, $\epsilon$ the modelling noise and $x$ the state vector. The forward model $F$ accounts for the atmosphere, instrument, surface and plume aerosol properties.





The optimal estimation uses an iterative approach that minimizes a cost function $\chi$ given by:

$$\chi = (x - x_a)^T S_a^{-1}(x - x_a) + (y - Kx)^T S_\epsilon^{-1}(y - Kx) \tag{5}$$

The Jacobian matrix of the problem $K$ is built using a Look-Up-Table (LUT) of radiative transfer simulations calculated
with MODTRAN (Berk et al., 2014) at each iteration. The state vector is limited to the aerosol plume parameters and composed
of the AOTs at $550\,\mathrm{nm}$ ($\delta_{550}$ and $\delta_{550}^*$), the median radius of the accumulation mode ($r_{median}$), the volume proportion of the
coarse mode in the size distribution ($V_{coarse}$) and the volume proportion of carbonaceous particles in the accumulation mode
($V_{soot}$). Associated with each parameter of $x$, a prior distribution is given by the state vector $x_a$ and the prior uncertainties in
the variance-covariance matrix $S_a$.

The matrix $S_\epsilon$ contains both the measurement uncertainties $S_y$ and the uncertainties of the model parameters that are not
retrieved, $S_b$. $S_b$ includes the different uncertainties due to: (i) the reconstruction of the surface reflectances under the plume
with the CNMF, (ii) the water vapor concentration in the atmosphere and (iii) the background aerosol visibility. The uncertain-
ties coming from the surface reflectances are obtained by calculating the spatial standard deviation for each wavelength of the
hyperspectral spectrum for pixels outside the spatial footprint of the plume (firstly visually identified). The uncertainties due
to the water vapor concentration are empirically fixed to 10%. Finally, the uncertainty of the background aerosol visibily is set
to $5\,\mathrm{km}$. The uncertainties of the unknown in the model $b$ are projected on the state space by approximating these uncertain-
ties to the first order thanks to $K_b$, the Jacobian matrix containing the partial derivatives associated to these parameters. The
observation uncertainties is given by:

$$S_\epsilon = S_y + K_b^T S_b^{-1} K_b \tag{6}$$

The cost function $\chi$ is minimized using the Levenberg-Marquardt algorithm. At each iteration the state vector $x_i$ is updated
by:

$$x_{i+1} = x_i + (K^T S_\epsilon^{-1} K + S_a^{-1} \times (1 + \gamma))^{-1}$$
$$\times [K^T S_\epsilon^{-1} K(y - Kx) + S_a^{-1}(x - x_a)]. \tag{7}$$

where $\gamma$ is a regularization term allowing to adjust the step size of the Levenberg-Marquardt algorithm.

The uncertainties associated to the posterior state vector $\hat{x}$ are contained in the variance-covariance matrix $\hat{S}$ given by:

$$\hat{S} = (S_a^{-1} + K^T S_\epsilon^{-1} K)^{-1} \tag{8}$$

The averaging kernel matrix is used for the purpose of error analysis. The averaging kernel matrix $A$ can be defined analyti-
cally as the product of the Jacobian matrix $K$ and the gain matrix $G$.

$$G = (K^T S_\epsilon^{-1} + S_a^{-1})^{-1} K^T S_\epsilon^{-1} \tag{9}$$





**Table 1.** Prior information fine mode aerosol median radius $r_m$, model type and volume fraction $V_c$. Std is the standard deviation of the lognormal mode.

|  | $r_m$ (std.) | type | $V_c$ (%) | $V_{soot}$ (%) |
|---|---|---|---|---|
| flaring | 0.065 (1.5) | soot | 52 | 100 |
| sinter plant | 0.13 (1.4) | sulfate + soot | 62 | 0 |
| coal-fired plant | 0.18 (1.5) | sulfate + soot | 52 | 0 |

The different terms of $A$ can also be defined as the partial derivative $\partial \hat{x}/\partial x^*$ representing the variation of the posterior state vector $\hat{x}$ with respect to the changes of the true state $x^*$. The diagonal elements of $A$ are the degrees of freedom (DOF) of the retrieved parameter. DOF evaluate the independence of each restitution to the prior constraint and range between 0 (totally depend on the prior vector) to 1 (totally dependent on the measurements).

### 3.4 Aerosol models

The size distribution of the aerosol models is bi-modal and includes an accumulation (fine) and a coarse mode. The refractive index of the accumulation mode is defined as an internal mixture between the refractive index of a scattering (sulfate) and an absorbing aerosol (soot) from the OPAC database (Hess et al., 1998). Prior information for the aerosol models is given in Table 1. The prior aerosol models and associated uncertainties are established following the literature review presented in section 2. The literature review proposes a range of aerodynamic diameters for the accumulation modes and partial information on the width of the size distribution. The aerodynamic diameter is set to $0.22 \, \mu m$, $0.67 \, \mu m$ and $0.60 \, \mu m$ for the flaring, sinter plan, and coal-fired plant respectively. Coarse mode for the sinter and coal-fired emissions is simulated as a dust-like non-spherical aerosol using an axis ratio of 2 (Mishchenko et al., 1997; Dubovik et al., 2002). The physical radius of the coarse mode is set to $0.5 \, \mu m$ (standard deviation of 2.0). The aerosol optical properties are simulated using the MOPSMAP *T-matrix* algorithm (Gasteiger and Wiegner, 2018). Prior estimate of the coarse mode fraction is based on reported particulate mass fraction. For flaring emission, as there is no reported value for $V_c$, we have fixed the prior value to the one of the coal-fired plant.

The simulated atmosphere contains a stack plume having a thickness of $100 \, m$ (Leoni et al., 2016) and a base at $50 \, m$ above the ground. The plume height is defined empirically however it has a negligible impact on the forward model simulations.

### 3.5 Retrieval of mass flow rate

The pixel-by-pixel columnar mass density $\Delta\Omega$ (in $g\,m^{-2}$) of the plume is given as the ratio between $\delta$ and the mass extinction efficiency $\alpha_{ext}$ (Hand and Malm, 2007). $\alpha_{ext}$ (at 550 nm wavelength) is a function of the retrieved aerosol model parameters given by the state vector $x$.

The uncertainties on $\Delta\Omega$ are estimated using the posterior uncertainties on $\delta$ and $\alpha_{ext}$. $\Delta\Omega$ is then used to estimate the mass flow rate at the point source. The different methods developed to estimate the mass flow rate from satellite imagery can





be classified in four families: inversion methods using Gaussian plumes (Bovensmann et al., 2010), the so-called source pixel methods (Jacob et al., 2016), the cross-sectional methods also called the cross sectional flux method (Tratt et al., 2011, 2014; Krings et al., 2011, 2013) and the integrated mass enhancement method (Varon et al., 2018, 2021; Frankenberg et al., 2016). The integrated mass enhancement (IME) method overcomes the shortcomings of the previously presented methods with respect to the configuration of the studied data, *i.e.* a plume image for which the knowledge of the wind comes from large mesh meteorological data and for which the knowledge of the stability of the atmosphere is unknown to us. IME for a given pixel $j$ of area $A_j$ is given by:

$$IME = \sum_{j=1}^{N} \Delta\Omega_j A_j, \tag{10}$$

The mass flow rate $Q$ (in $\mathrm{g\,s^{-1}}$) is the ratio between the IME and a characteristic resident time of the particles in the detected plume. The resident time can be expressed as the ratio between an effective wind speed $U_{eff}$ and a characteristic length of the plume $L$ (Frankenberg et al., 2016; Varon et al., 2018), leading to:

$$Q = \frac{U_{eff}}{L} \times IME, \tag{11}$$

As stated by Varon et al. (2018), $U_{eff}$ and $L$ must be viewed as operational parameters to be related to the observed wind speed and plume extent. The definition of $L$ will influence the relationship between $U_{eff}$ and the 10-m wind. $L$ is usually defined as equal to the square root of the area of the detected pixels. In the case of constant direction propagation over time, $L$ can be chosen as equal to the length of a study area in the plume propagation direction and $U_{eff}$ as an average wind speed. In this configuration, the IME calculation is equivalent to the calculation of an average cross sectional flux.

The flow rate uncertainties $\delta Q$ are estimated using the relative uncertainties on $U_{eff}$ and $\Delta\Omega$. $U_{eff}$ and its relative error are estimated from the ensemble model of ERA-5 re-analysis.

## 4 Retrieval of aerosol optical properties in industrial stack plumes

### 4.1 Gas flaring

In the case of the Hassi Messaoud gas flare, the surface reflectances are easily reconstructed by CNMF due to a homogeneous ground surface. Moreover, the high contrast between the plume and the ground favors the detection of the plume.

The DOFs associated with the retrieved AOT and median radius, as well as the number of iterations in the LM algorithm are used to detect the spatial footprint of the plume. The DOF on the AOT is equal to 0.99 on average and the values are spatially homogeneous in the plume. Applying a threshold of 0.5 on the median radius DOF (Figure 5a) highlights the footprint of the plume, except near the source. The mask obtained with DOF>0.5 keeps the pixels whose retrievals are more than 50% independent of the prior constraint.



The number of iterations of LM algorithm also gives a good proxy for the plume footprint. The visual comparison between the color composition (Figure 3) and the number of iterations of LM algorithm (Figure 5b) indicates that most of the plume is found within 10 iterations. We note that at the source level, the LM algorithm does not converge probably due to the flame thermal emission. Indeed, the temperature of this flame is estimated to $1750\,\mathrm{K}$ according to Skytruth[1] and leads to a significant emission in the SWIR that is not accounted for in our model. Finally, the combined plume mask based on DOF and iteration number (Figure 5c) corresponds well to the visual inspection of the corresponding color composition (Figure 3).

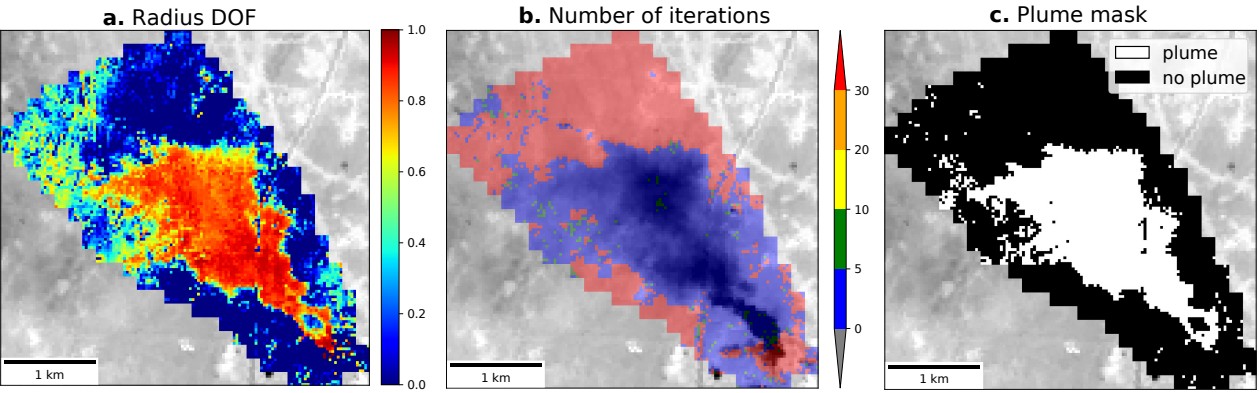

**Figure 5.** OEM results on the flaring plume: (a) median radius DOF, (b) Number of iterations in the LM algorithm, (c) combined mask

The initial value of AOT is set to 0.5 with an uncertainty of 1.0. The retrieved AOT map (Figure 6a) reflects the plume structure and shows several maxima in the plume, near the point source and downwind. The mean plume AOT is equal to 0.27 (see summary Table 3) to which is associated a mean statistical uncertainty equal to 0.011.

Some artifacts that are due to CNMF recontruction are located near the road and correspond to sandy structures that have moved between the PRISMA and the S2/MSI acquisitions.

The retrieved radii (Figure 6b) are rather homogeneous with a spatial variation of $0.03\,\mathrm{\mu m}$ and on average equal to $0.12\,\mathrm{\mu m}$ with a statistical uncertainty of $0.02\,\mathrm{\mu m}$. The final $V_c$ (Figure 6c) converges to 46 % with a spatial standard deviation of 13%.

## 4.2 Sinter plant

In the case of the sinter plant, one must pay attention to the geometry of illumination of the scene. Indeed, we can observe in Figure 2 that the plume propagation direction is $20°$ with respect to the geographical North. The solar azimuth angle $\theta_s$ is $144°$. So the angular difference between the direction of illumination of the sun and the direction of the plume is $56°$. For a plume having a vertical extent of $\approx 500\,\mathrm{m}$, the distance between the crossing points of the downward flux and the upward flux is almost equal to the widest part of the plume. Consequently, an additional AOT associated to $E_d$ (see section 3.2 and discussion section hereafter) is also retrieved. Moreover, a variable fraction of soot particles in the accumulation mode (volume fraction $V_{soot}$) is introduced in the state vector to better fit the aerosol hyperspectral signal.

---

[1]Skytruth website : https://skytruth.org/flaring/





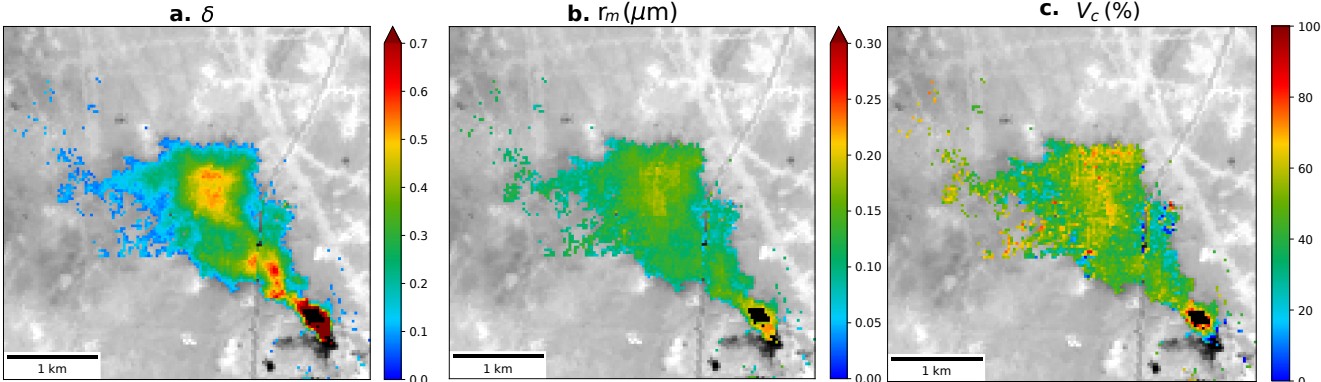

**Figure 6.** Estimation of (a) aerosol optical thickness at $550\,\mathrm{nm}$, (b) fine mode median radius and (c) coarse mode fraction in the flaring plume.

The ascending part of the plume appears, as in the case of Hassi Messaoud, to be the densest (Figure 7) The plume puffs are perceptible through AOT local maxima.

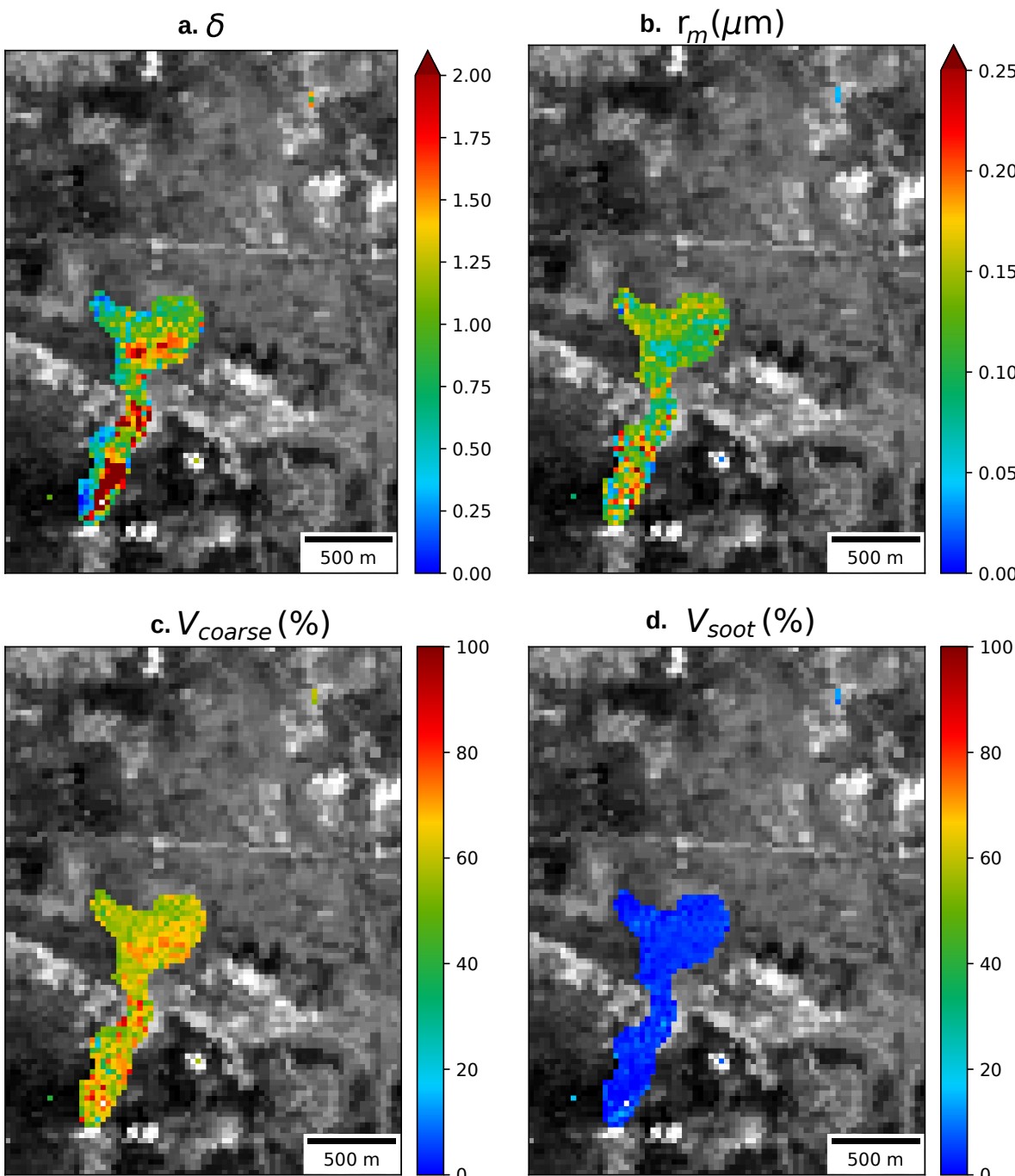

**Figure 7.** Estimation of (a) aerosol optical thickness at 550 nm, (b) fine mode median radius and (c) coarse mode fraction and (d) volume of soot in the sinter plant plume.





Finally, the other retrieved parameters of the state vector are spatially homogeneous. The median radius of the accumulation mode is on average equal to $0.11 \pm 0.02$ $\mu$m in the pixels present in the plume mask. The statistical uncertainties associated with these radii are equal to $0.05$ $\mu$m. The volume proportion of the coarse mode is 59% (with a statistical uncertainty of 14%), being 20% higher than the measurements found in the literature.

### 4.3 Coal-fired plant

For the coal-fired plant case study, the plume mask detection fails to recover the entire plume (Figure 8). Only the densest part of the plume near the source, as well as another area of vegetation with dark and homogeneous surface reflectances (Figure 8) are identified. The partial detection can be due to a poor reconstruction of the vegetated soils in the visible part of the spectrum because of the growth of the vegetation between the PRISMA and S2/MSI images (8 days). Further downstream, an artifact due to the sunglint on a water retention bassin can observed.

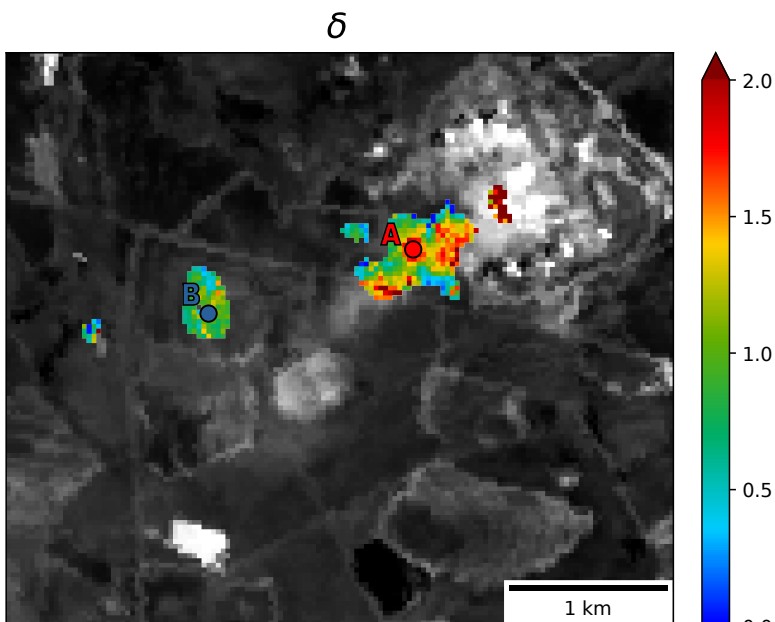

**Figure 8.** Optical thickness map at 550 nm obtained from the OEM. A local study of the OEM retrievals is performed at points A and B.

To illustrate the variability in the retrieval conditions, we have selected two areas of 3×3 pixels in each detected zone (A and B in Figure 8). Point A is located in a dense part of the plume, while point B is located downwind. Both points are located over vegetated areas. Retrieved AOT at point A is 1.8 while it is 0.8 further downwind at point B (see Table 2). The sensitivity of the retrieval model to the spectral variations induced by the different physical parameters is proportional to the particle concentration in the plume. We first observe that the degrees of freedom associated with the radius are higher for point A

(0.65) than for point B (0.51), indicating that the point B retrieved radius is strongly constrained by its prior information than





at point A. The radius is equal to $0.20\,\mu m$, which is relatively close to the prior value, while the retrieved radius for point A is equal to $0.10\,\mu m$ Finally, it is interesting to note that the retrieved AOT associated with the descending solar flux ($\delta^*$) is much lower at point B than point A due to its shifted position with respect to the propagation axis of the plume.

**Table 2.** Aerosol optical parameters in the coal-fired plant plume at selected location A and B (see Figure 8).

|   | radius (µm) | $\delta$ | $\delta^*$ | $V_{soot}(\%)$ | $V_{coarse}(\%)$ | $DOF_{radius}$ |
|---|---|---|---|---|---|---|
| A | 0.10 | 1.86 | 2.05 | 9.25 | 84 | 0.65 |
| B | 0.20 | 0.80 | 0.12 | 13.04 | 67 | 0.51 |

## 5 Mass flow rate

The surface mass concentration $\Delta\Omega$ (Figure 9) is estimated using the retrieved AOT and the mass extinction efficiency (see section 3.5). The detection limit (borders of the plumes) is around $0.1\,\mathrm{g\,m^{-2}}$. The surface mass concentration is up to $2\,\mathrm{g\,m^{-2}}$ in the case of the coal-fired plant plume. The surface mass calculated in the densest areas of plumes corresponds to an atmospheric concentration between $1\,\mathrm{mg\,m^{-3}}$ and about $10\,\mathrm{mg\,m^{-3}}$ for a plume vertical extent of 100m. The mass flow rate is estimated in selected parts of the plumes after visual inspection (red rectangles in Figure 9). The selected areas are outside the rising part

of the plume. The effective wind speed $U_{eff}$ is equal to $7\,\mathrm{m\,s^{-1}}$, $3\,\mathrm{m\,s^{-1}}$ and $2\,\mathrm{m\,s^{-1}}$ for the flaring emission, sinter plant and coal-fired plant, respectively. The flow rate is splitted into the fine and coarse mode components (Table 3). The total estimated flow rate varies from $511\,\mathrm{g\,s^{-1}}$ for the flaring emission to $1401\,\mathrm{g\,s^{-1}}$ for the coal-fired plant emission.

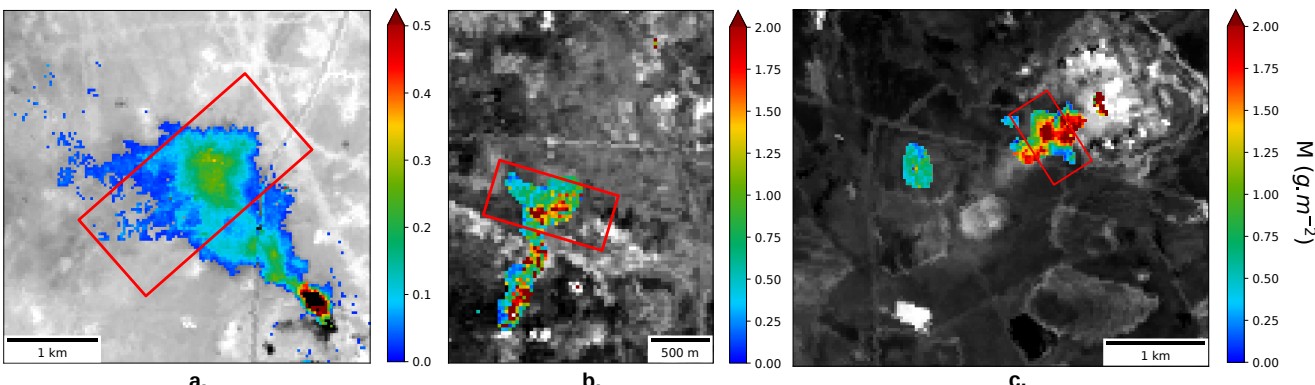

**Figure 9.** Plume surface mass (in $g \cdot m^{-2}$) retrieved for (a) flaring emission, (b) sinter plant emission, and (c) coal-fired plant emission. Red rectangles are the selected area for mass flow rate estimation.





**Table 3.** Average retrieved plume aerosol properties.The aerosol optical thickness $\delta$ and the mass extinction efficiency $\alpha_{ext}$ are given at 550 nm.

| | retrieved aerosol parameters | | |
|---|---|---|---|
| | flaring | sinter | coal-fired |
| $r_{median}$ (µm) | 0.12 | 0.11 | 0.10 |
| $\delta$ (no unit) | 0.27 | 0.94 | 1.27 |
| $V_{coarse}$ (%) | 46 | 59 | 81 |
| $V_{soot}$ (%) | 100.0 | 3.6 | 10.4 |
| $\alpha_{ext}$ (m$^2$ g$^{-1}$) | 3.07 | 1.20 | 0.99 |
| mass flow rate (g s$^{-1}$) | | | |
| Fine mode | 394 | 383 | 131 |
| Coarse mode | 446 | 965 | 926 |
| Total | 840 | 1348 | 1057 |

# 6 Discussion

## 6.1 Uncertainty analysis

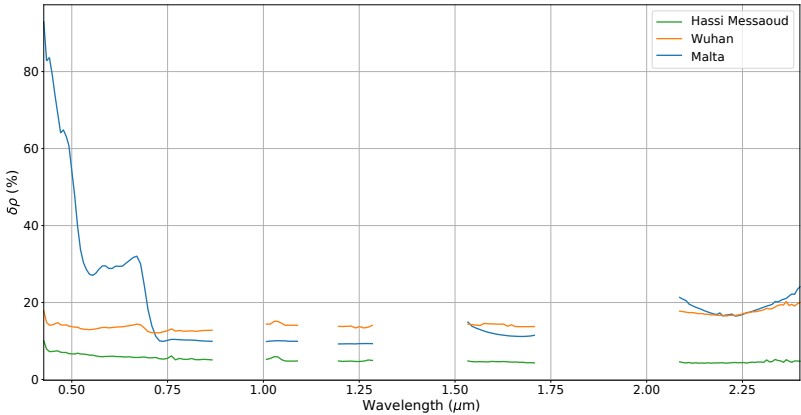

**Figure 10.** Relative error on the surface reflectance reconstruction for the flaring site (blue), sinter plant (orange) and coal-fired plant plant (green).

The surface reflectance reconstruction below the plume is affected by (i) the co-registration process, (ii) the time lag between PRISMA and S2/MSI acquisitions (iii) and the endmembers used for the CNMF. For heterogeneous scenes, like the sinter plant and the coal-fired plant, the endmembers are less representative than for homogeneous scenes like the flaring site. Although the surface reflectances are high around the flaring site ($\approx 0.1$), the associated error is 2 times lower than for the 2 others studied cases (Figure 10). The vegetation growth between the PRISMA and S2/MSI acquisition in the case of the coal-fired plant has





a dractic impact on the reflectance reconstruction (Figure 10). The relative error on the surface reflectance reaches 60% at 550 nm for this particular cases.

The error associated to the surface reflectance estimation and to the aerosol properties retrieval process are used to esti-mated the uncertainties on the plume surface mass estimate. The surface mass uncertainties ($\delta\Delta\Omega$) is estimated by using the uncertainties on the AOT (hereinafter $\delta(AOT)$) and on the mass extinction efficiency ($\delta\alpha_{ext}$).

$$\delta\Delta\Omega = \sqrt{\left(\frac{\delta(AOT)}{AOT}\right)^2 + \left(\frac{\delta\alpha_{ext}}{\alpha_{ext}}\right)^2} \times \Delta\Omega \quad (12)$$

and $\delta\alpha_{ext}$ can be decomposed into:

$$\delta\alpha_{ext} = \sqrt{\sum_{i=1}^{n}\left(\frac{\partial\alpha_{ext}}{\partial\hat{x}_i}\right)^2 \hat{S}_{i,i}} \times \alpha_{ext} \quad (13)$$

where $i$ represents the values of all the parameters of the posterior state vector $\hat{x}$ except AOT and $\hat{S}_{i,i}$ the variances of each of these parameters present on the diagonal of the variance-covariance matrix $\hat{S}$. The $\hat{S}$ matrix also contains the surface reflectance
CNMF error. The estimated uncertainties on the retrieved parameters are given in Table 4.

**Table 4.** Uncertainties on the retrieved aerosol parameters.

|  | flaring | sinter | coal-fired |
|---|---|---|---|
| $\delta\Delta\Omega$ (mg m$^{-2}$) | 0.21 | 4.78 | 3.39 |
| $\delta\alpha_{ext}$ (m$^2$ g$^{-1}$) | 0.62 | 0.47 | 0.27 |
| $\delta r_m$ (μm) | 0.02 | 0.05 | 0.02 |
| $\delta(AOT)$ (no unit) | 0.01 | 0.35 | 0.08 |
| $\delta V_c$ (%) | 12.18 | 14.03 | 3.82 |
| $\delta V_{soot}$ (%) | - | 6.15 | 2.64 |

The contribution of the error on the retrieved parameters to the surface mass uncertainty depends on the case study. When the surface reflectances are poorly reconstructed (sinter and coal-fired plant) the AOT error contribution contribution is around 40% while the contribution is 7% for the flaring site (Table 5). However the error associated to the retrieval of the coarse mode fraction in the case of the flaring emission drastically impacts the error on the surface mass concentration (75%). The error
associated to the estimation of the soot fraction in the accumulation mode has rather a weak impact on final error on the surface mass concentration. Indeed the soot fraction is rather low (see Table 3) for the sinter and coal-fired plant plumes.

The error on the flow rate is due to the cumulative error on the effective wind speed and the surface mass concentration (see Table 5, bottom part). The wind uncertainties are equal to $1\,\mathrm{m\,s^{-1}}$, $0.5\,\mathrm{m\,s^{-1}}$ and $0.5\,\mathrm{m\,s^{-1}}$ for the flaring emission, the sinter and coal-fired plant, respectively. The wind speed uncertainty represents a substantial contribution to flow rate variance, which





could be decreased thanks to a better knowledge of local atmospheric conditions. In the case of the sinter plant, the variance in the flow rate estimate is still largely dominated by the error on large error on the surface mass concentration (see Table 4).

Another source of error on the estimation of the flow rate comes from the definition of $U_{eff}$ in the IME method. The relationship between $U_{eff}$ and the 10-m wind speed depends on the measurements condition, on the method used to estimate the flow rate and on the intrument specifications (Varon et al., 2018). Based on large-eddy simulations for methane plumes, an

underestimation of the actual flow rate by 30 to 50% can be expected (Nesme et al., 2021; Varon et al., 2021).

**Table 5.** Relative contribution (in %) of parameters uncertainty to the surface mass variance and to the flow rate variance.

| | contribution to surface mass variance | | |
| --- | --- | --- | --- |
| | flaring | sinter | coal-fired |
| $r_m$ (%) | 18 | 24 | 35 |
| $\delta$ (%) | 7 | 46 | 40 |
| $V_c$ (%) | 75 | 28 | 22 |
| $V_{soot}$ (%) | - | 2 | 3 |
| | contribution to flow rate variance | | |
| Wind (%) | 34 | 7 | 40 |
| $\Delta\Omega$ (%) | 66 | 93 | 60 |

## 6.2 Concentration and flow rate

As we have selected remarkable plumes, the flow rate must be seen as top estimations. The total flow rate calculated for each site is in the upper range of expected values, being about $1\,\mathrm{kg\,s^{-1}}$. The sites are large scale facilities in countries having legislation less restrictive than the European standards. For Hassi Messaoud flaring site, the flares emit 0.10 billion cubic meters

(BCM) of gas in 2021 according to SkyTruth. Applying the emission factor of Caseiro et al. (2020), the corresponding annual emission of fine particles is $10^8\,\mathrm{g}$. Assuming that these flares emit continuously over the year, the average flow rate associated with a flare is $3\,\mathrm{g\,s^{-1}}$. The fine mode flow rate associated with one flare is $98.5\,\mathrm{g\,s^{-1}}$ (the plume is generated by 4 flares), being about 2 orders of magnitude greater than the average flow rate. By analyzing 130 of S2-A and S2-B images over Hassi Messaoud for which the flares were visible, we have detected only 5 plumes having the same size or even larger. Consequently,

the case study of Hassi Messaoud is probably an uncommom phenomenom.

The impact on the air quality in the surrounding of the facilities will drastically depends on the ventilation of the plume and how it vertically disperses. The detection limit of $0.1\,\mathrm{g/m^2}$ for the visible plume corresponds to an atmospheric concentration of $200\,\mathrm{\mu g/m^3}$ for a plume vertical extent of $500\,\mathrm{m}$. The atmospheric concentration is about one order of magnitude above the $50\,\mathrm{\mu g/m^3}$ limit of the EU Directire 2008/50/EC that must not be exceeded more than 35 times during a calendar year. There are

several limitations (low revisit time, cloud cover, ...) for further investigations of the applicability of the proposed framework to monitor the air quality around the facilities. However, high spatial resolution observations can provide unique information for understanding the impact of industrial emissions on the environment.





# 7    Conclusion

We propose an inversion framework to retrieve instantaneous particulate matter emission by industrial stacks using hyper-
spectral PRISMA satellite images. Aerosol plume satellite retrieval over continental surfaces is a challenge and we had to
implement different steps to unravel the impact of the underlying surface and the impact of the particle size, concentration and
type on the satellite signal. At first, the fusion algorithm with operational S2/MSI images provides an estimate of the surface
reflectance and its uncertainties below the plume. The radiative impact of the plume on a background atmosphere is then sim-
ulated for varying plume particle median radius, aerosol optical thickness and volume proportion of soot also considering the
geometrical conditions of the scene. The uncertainties associated to the surface reflectance estimation are propagated to the
final aerosol solution using the OEM formalism. The use of OEM allows to retrieve a plume mask and the associated aerosol
properties. Setting up the prior aerosol model for each type of emission was achieved based on available but scarce literature
review. The inversion was tested over 3 types of industries: a coal-fired plant, a sinter plant and an oil flaring site. In most of
the cases not all the plume area can be investigated and the mass flow rate was estimated by selecting limited portions of the
plume. Moreover, the relationship between actual wind speed and IME effective wind speed would required further investi-
gations considering the uncertainties on the retrieved surface mass as well as the plume dynamic. Nevertheless, the retrieved
aerosol physical characteristics and the estimated instantaneous emission flow rate are within the expected range for each type
of emission.

The synergy between PRISMA and S2/MSI could be improved in several ways. Using a time series of S2/MSI rather than
the nearest image in time could improve the estimation of the surface reflectance variability. Moreover, the aerosol model as
retrieved using the hyperspectral imager could be prescribed to S2/MSI with the aim of providing regular acquisition of the
plume time evolution. Lastly, going down to a 10 m spatial resolution could improve the detection of narrow plumes provided
that the surface is rather homogeneous.

A comprehensive validation exercise of the retrieved parameters would require in situ measurement of the mass float rate
by aerosol size fraction, aerosol size distribution and chemical composition within the plume as well as the plume extent.
However, due to the large difficulties of accessing industrial infrastructures, the validation of the retrieved mass flow rate or
aerosol properties remains virtually impossible for our case studies. At least a consistency check could be tested between soot
emission by flares based on the stack emission temperature and the retrieved soot flow rate. And as a perspective, the use of
high spatial resolution satellite for aerosol retrievals on industrial sites is also promising for the improvement of top-down
emission inventories as the number of hyper spectral satellite missions is expected to increase in the near future.

*Data availability.* PRISMA data are available to download at http://prisma.asi.it/missionselect/ (Italian Space Agency, 2023, a sign-in is
required). Aerosol retrievals are available upon request. All Sentinel-2 satellite data used for this study are publicly available through the
Copernicus Open Access Hub (https://scihub.copernicus.eu/, ESA, 2023).



*Author contributions.* GC performed data analysis and contribute to the manuscrit. PYF and JFL supervised data analysis, and contributed to the manuscript.

*Competing interests.* The contact author has declared that none of the authors has any competing interests.

*Financial support.* This research has been supported by CNES under project named IMHYS.

*Acknowledgements.* The authors would like to thank the Italian Space Agency for the PRISMA data used in this work.




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
