# Peer review of "Quantifying particulate matter optical properties and flow rate in industrial stack plumes from PRISMA hyperspectral imager"

_Atmospheric Measurement Techniques, 2023_

## Referee Comment (RC1)

**General Comments:**

The study "Quantifying particulate matter optical properties and flow rate in industrial stack plumes from PRISMA hyperspectral imager" by Calassou et al. addresses the issue of industrial stacks plumes retrieval by means of PRISMA satellite data. The topic is scientifically relevant and the paper is well structured and written. Thus, I recommend the manuscript for publication after addressing the following points and clarifications.

1- Is it the first time that PRISMA data have been used for this type of research? If yes it would be valuable to highlight that, if not you should provide references in literature.
2- Have you checked if other studies on the same test sites have been conducted in literature?
3- Have you considered the issue of clouds influence? Are the images you've taken into account (both PRISMA and L2/MSI) free of clouds?
4- Have you considered the possibility to provide an independent validation of the presented results? Maybe with existent satellite products/derived satellite products/other ground truth sources; in line 354-357 you mention that the validation with in situ measurements remains virtually impossible. Have you investigated if there are satellite products that provide the same parameters you estimated?

**Specific Comments:**

- Line 100: do you mean only detection or also retrieval?
- Eq.1 -> Is there a reference in literature?
- Eq.2 -> Is there a reference in literature?
- Line 128: Is there a reference in literature to the COMANCHE software?
- Eq.3 -> Is there a reference in literature? -> Could you clarify why there is no direct component of the solar irradiance related to $\delta$?
- Line 165: "the uncertainties due to the water vapor concentration are empirically fixed to 10%" -> Is there a reference in literature for the uncertainty due to the water vapor concentration?
- Line 166: "the uncertainties of the background aerosol visibility is set to 5 km" -> -> Is there a reference in literature for the uncertainty of the background aerosol visibility?
- Line 227: LM acronym has not been defined previously.
- Line 236: "the combined plume mask based on DOF and iteration number (Figure 5c)" -> Could you clarify how DOF and iteration number are combined to get the plume mask (ex.: "pixels corresponding to DOF threshold of ... and?/or? pixels corresponding to number iteration of ... are taken into account...")
- Caption of Figure 5: OEM acronym has not been defined previously.
- Line 238: "The initial value of AOT is set to 0.5 with an uncertainty of 1.0. The retrieved AOT map (Figure 6a) reflects[...]" -> Could you mention/briefly recall the process to retrieve the parameters in Figure 6 (how to obtain Figure 6 from Figure 5 basically). Is the process mentioned in line 192-193 ("The aerosol optical properties are simulated using the MOPSMAP T-matrix algorithm (Gasteiger and Wiegner, 2018)") ?
- Line 256: "The median radius of the accumulation mode is on average equal to $0.11 \pm 0.02$ µm in the pixels present in the plume mask." -> The plume mask (as well as the DOF and Number

of iterations) have been shown only for case 1 (Gas flaring, Fig. 5), I suppose the procedure is exactly the same for all the sites, is that correct?

- Line 262-263: "The partial detection can be due to a poor reconstruction of the vegetated soils[...]" -> May be the partial detection caused also by the fact that the plume may be less thick/more dispersed in that areas?
- Line 263-264: "Further downstream, an artifact due to the sunglint on a water retention bassin can observed" -> Where the artifact/water retention bassin is located in the map?
- Figure 8: May the point B be a false positive?
- Line 290-291: "The relative error on the surface reflectance reaches 60% at 550nm for this particular cases" -> Could you specify which case? In the previous sentence the coal-fired plant has been mentioned while from Figure 10 it seems that the flaring site exhibits a relative error of 60% at 550nm. Could you clarify that and check the correspondence between text and Figure 10 from 285 to 291 lines?
- Line 292: "The error associated to the surface reflectance estimation and to the aerosol properties retrieval process"-> Could you provide a reference in the text on where in the manuscript the error associated with the aerosol properties retrieval process is computed? Is the 3.3 section?
- Line 330: "There are several limitations (low revisit time, cloud cover, ...)"-> Could you specify limitations of the "applicability of the proposed framework to monitor the air quality around the facilities"

**Technical Comments:**

- Line 26: stationnary -> stationary
- Line 30: developped -> developed
- Line 47: impletation -> implementation
- Line 50: betwen -> between
- Line 116: space after the point "[...] m$^2$).For a flat, [...]"
- Line 165: visibily -> visibility?
- Line 271: "The radius is equal to 0.20 μm" -> Do you mean the radius of the point B? Could you specify?
- Line 290: "dractic"->drastic
- Line 302: "contribution contribution"-> contribution
- Line 311:" dominated by the error on large error"-> Could you review this part of the sentence?

---

## Author Comment (AC1)

**Dear Reviewer,**

**We would like to thank you for your remarks and comments, which have helped us to improve the manuscript. Please find below our answers (marked in bold blue color).**

**General Comments:**

The study "Quantifying particulate matter optical properties and flow rate in industrial stack plumes from PRISMA hyperspectral imager" by Calassou et al. addresses the issue of industrial stacks plumes retrieval by means of PRISMA satellite data. The topic is scientifically relevant and the paper is well structured and written. I recommend the manuscript for publication after addressing the following points and clarifications.

- Is it the first time that PRISMA data have been used for this type of research? If yes it would be valuable to highlight that, if not you should provide references in literature.

**To the best of our knowledge, this is the first study published on this subject.**

- Have you checked if other studies on the same test sites have been conducted in literature?

**Only the site of Hassi Messaoud have been investigated for CH4 emission: Lauvaux et al. 2022) already in the text and we have added Varon et al. (2021). L90.**

- Have you considered the issue of clouds influence? Are the images you've taken into account (both PRISMA and L2/MSI) free of clouds?

**We have selected cloud free pictures and the cloud influence is not accounted for in the present article.**

- Have you considered the possibility to provide an independent validation of the presented results? Maybe with existent satellite products/derived satellite products/other ground truth sources; in line 354-357 you mention that the validation with in situ measurements remains virtually impossible. Have you investigated if there are satellite products that provide the same parameters you estimated?

**Yes, we have considered this possibility, but so far we haven't found any equivalent products at the same spatial resolution and on the same study cases as ours. Ground-based optical measurements (eg DOAS or scanning lidar system) might be valuable for vicarious validation in future researches.**

**Specific Comments:**
- Line 100: do you mean only detection or also retrieval?

**Detect and quantify methane emission. Added to the text.**

- 1 -> Is there a reference in literature?
- 2 -> Is there a reference in literature?

**Yes. We have added the following references : Liou (2002), Vermote et al. (1997) and Kaufman et al. 1997.  Line 124.**

- Line 128: Is there a reference in literature to the COMANCHE software?

**Yes. Poutier et al. 2002. Added in the text.**

- 3 -> Is there a reference in literature? -> Could you clarify why there is no direct component of the solar irradiance related to δ?

**No reference. It's a new formalism.  As mentioned in the text, δ is the AOT for the upward irradiance so there no direct component of the solar irradiance.   δ\* is the AOT for the downward irradiance. It's now clarified in the text. Line 135.**

- Line 165: "the uncertainties due to the water vapor concentration are empirically fixed to 10%" -> Is there a reference in literature for the uncertainty due to the water vapor concentration?

  **And**

- Line 166: "the uncertainties of the background aerosol visibility is set to 5 km" -> -> Is there a reference in literature for the uncertainty of the background aerosol visibility?

**Uncertainties in the estimation of water vapor range from 1 to 15%. For a PRISMA-type instrument, a value of 10% error on this parameter seems relevant. In the same way as for water vapour, the visibility error estimate is around 5 km.**
**We have explained this point in the text and added the following references:**

**Yang, H.; Zhang, L.; Ong, C.; Rodger, A.; Liu, J.; Sun, X.; Zhang, H.; Jian, X.; Tong, Q. Improved Aerosol Optical Thickness, Columnar Water Vapor, and Surface Reflectance Retrieval from Combined CASI and SASI Airborne Hyperspectral Sensors. *Remote Sens.* 2017, *9*, 217. https://doi.org/10.3390/rs9030217**

**Andrew Rodger,SODA: A new method of in-scene atmospheric water vapor estimation and post-flight spectral recalibration for hyperspectral sensors: Application to the HyMap sensor at two locations,Remote Sensing of Environment,Volume 115, Issue 2,2011,Pages 536-547,ISSN 0034-4257,https://doi.org/10.1016/j.rse.2010.09.022.**

**Nitin Bhatia, Alfred Stein, Ils Reusen & Valentyn A. Tolpekin (2018) An optimization approach to estimate and calibrate column water vapour for hyperspectral airborne data, International Journal of Remote Sensing, 39:8, 2480-2505, DOI: 10.1080/01431161.2018.1425565**

- Line 227: LM acronym has not been defined previously.

**This has been done on Line 170**

- Line 236: "the combined plume mask based on DOF and iteration number (Figure 5c)" -> Could you clarify how DOF and iteration number are combined to get the plume mask (ex.: "pixels corresponding to DOF threshold of ... and?/or? pixels corresponding to number iteration of ... are taken into account...")

**Yes. The resulting mask is the product of both masks, ie. DOF>0.5 and Number of iteration<10. We have clarified this point in the text.**

- Caption of Figure 5: OEM acronym has not been defined previously.

**This has been done on Line 148**

- Line 238: "The initial value of AOT is set to 0.5 with an uncertainty of 1.0. The retrieved AOT map (Figure 6a) reflects[...]" -> Could you mention/briefly recall the process to retrieve the parameters in Figure 6 (how to obtain Figure 6 from Figure 5 basically). Is the process mentioned in line 192-193 ("The aerosol optical properties are simulated using the MOPSMAP T-matrix algorithm (Gasteiger and Wiegner, 2018)") ?

**We recall briefly the procedure L240:**
**The AOT is a component of the state vector in the optimal estimation procedure. Its value is retrieved by minimizing the cost function that integrates the look-up tables computed using the direct radiative transfer model and the aerosol optical properties (section 3.3 and 3.4).**

- Line 256: "The median radius of the accumulation mode is on average equal to $0.11 \pm 0.02$ µm in the pixels present in the plume mask." -> The plume mask (as well as the DOF and Number of iterations) have been shown only for case 1 (Gas flaring, Fig. 5), I suppose the procedure is exactly the same for all the sites, is that correct?

**Yes. Now explicitly given in the text. L252.**

- Line 262-263: "The partial detection can be due to a poor reconstruction of the vegetated soils[...]" -> May be the partial detection caused also by the fact that the plume may be less thick/more dispersed in that areas?

**Correct. For every pixel where the surface reconstruction error is high there is a low sensitivity to aerosol optical depth so the detection is not possible. We have mentioned this point in the text. Line 270.**

- Line 263-264: "Further downstream, an artifact due to the sunglint on a water retention bassin can observed" -> Where the artifact/water retention bassin is located in the map?

**Actually this sentence is awkward. The sunglint is not clearly visible. We have removed this sentence.**

- Figure 8: May the point B be a false positive?

**No. Point B is clearly identified as a retrieval and the AOT is out of the noise.**

- Line 290-291: "The relative error on the surface reflectance reaches 60% at 550nm for this particular cases" -> Could you specify which case? In the previous sentence the coal-fired plant has been mentioned while from Figure 10 it seems that the flaring site exhibits a relative error of 60% at 550nm. Could you clarify that and check the correspondence between text and Figure 10 from 285 to 291 lines?

**The corresponding text has been clarified. The 60% error at 500 nm is for the coal-fired plant. Figure 10 legend is updated.**

- Line 292: "The error associated to the surface reflectance estimation and to the aerosol properties retrieval process"-> Could you provide a reference in the text on where in the manuscript the error associated with the aerosol properties retrieval process is computed? Is the 3.3 section?

**Yes. The error associated with the aerosol properties retrieval is given by the error matrix Ŝ in equation 8. We add this point in the text. Line 301.**

- Line 330: "There are several limitations (low revisit time, cloud cover, ...)"-> Could you specify limitations of the "applicability of the proposed framework to monitor the air quality around the facilities"

**You are right, the two last sentence are unclear and probably clumsy. We have rephrased them. Line 340.**

**"Although the proposed method is not dedicated to the monitoring of air quality around the facilities, high spatial resolution observations of plume transport can provide unique information for understanding the impact of industrial emissions. The main limitation for an operational survey of plume emissions are the reduced amount of observations due to the cloud occurrence and the revisit time of the satellite."**

**Technical Comments:**

**Thank you for your careful proofreading. We have proofread the text and corrected the typos.**

- Line 26: stationnary -> stationary
- Line 30: developped -> developed
- Line 47: impletation -> implementation
- Line 50: betwen -> between
- Line 116: space after the point "[...] m$^2$).For a flat, [...]"
- Line 165: visibily -> visibility?
- Line 271: "The radius is equal to 0.20 µm" -> Do you mean the radius of the point B? Could you specify?

**Correct. Aerosol radius at point B.**

- Line 290: "dractic"->drastic
- Line 302: "contribution contribution"-> contribution
- Line 311:" dominated by the error on large error"-> Could you review this part of the sentence?

---

## Author Comment (AC2)

**Dear Reviewer,**

**We would like to thank you for your remarks and comments, which have helped us to improve the manuscript. Please find below our answers (marked in bold blue color).**

The reviewed manuscript focuses on the development of an inversion framework using spaceborne PRISMA hyperspectral imager to estimate particulate matter emissions from industrial stacks. It addresses the challenges associated with aerosol plume retrieval over continental surfaces, considering, e.g. surface interference and particle characteristics. The study presents a methodology that incorporates surface reflectance estimation, radiative simulations, and an optimal estimation framework to retrieve aerosol properties and emission flow rates. The reliable retrieval of aerosol properties is of vital importance for correct estimate of the ecological footprint of a given factory or country as well as for the public health, so any effort improving our knowledge is laudable, and I think that the manuscript is topical. It is well organized and the presentation is clear. There are few things in the methodological part, though, which I'd like to see improved or clarified, so I've selected "major revision", but the changes I propose below are easy to implement.

**General comments:**

- The authors did a good job by presenting their model and by providing the optimal estimation formalism. But, the method lacks such an important validation step as a self-consistency study, which would both assure the reliability of the proposed method and estimate its uncertainty. Let me explain: if one has a forward model and the inversion algorithm for some instrument, one should be able to take the initial state of the atmosphere (some synthetic image or a retrieved one, with some prescribed aerosol size distribution), name it a "reference", run the forward simulation and obtain the "would be" multispectral image. Then, this information should be fed to an inversion algorithm, which should be run for several times, each time starting with a different guess state. The retrieved aerosol properties should be compared with the reference ones and the biases and root-mean-square values should be estimated. Presumably, a purely theoretical noise-free retrieval will give the result close to the reference. This will tell us about the reliability of the method. In the next step, the simulated radiances should be modified using some realistic noise and the retrieval should be repeated. The biases and r.m.s. values got at this step would tell us about the real uncertainty of the data presented in the manuscript and obtained from real observations using the suggested method. I saw the discussion of the uncertainties in Section 3.3, but as far as I understand, this does not clarify the self-consistency issues outlined above.

**We perfectly understand your point. However we believe that the optimal estimation method is widely used remote sensing application and the link between the measurement noise and the retrieval uncertainties in such a formalism is well controlled (eg. Rodgers and successive applications). We agree that the OEM is sensitive to the prior information and that's why we paid a particular attention to the definition of the prior state vector using literature review. Moreover we control the convergence of the problem using the DOF and iteration number of**

**the Levenberg-Marquartd algorithm. The main advantage of the OEM is to have a diagnostic of the posterior error on the retrieved values.**

- Another validation step I did not find in the manuscript is the comparison with existing observations. All three test cases happened in the period when the CALIOP/CALIPSO lidar was still operable, and the corresponding aerosol information retrieved from CALIOP observations should be available. Another space-borne lidar observing the Earth in the same period was ALADIN/Aeolus, but the comparison of PRISMA retrievals with Aeolus L2A optical data might be difficult due to its long averaging along the track (in this case, the apples-to-apples comparison would require averaging the PRISMA data along the same 87km length). I understand that a good overlapping for a given date is not guaranteed, but I believe that one can pick up another day, which would give the same image as in the figures provided in the manuscript, and at the same time the image would be overlapped with the CALIPSO track. Such a comparison would be a good validation both for the method and for the measurement itself. I do not suggest to redo the whole study, but at least one collocation and one comparison plot for one site are needed.

**We thank you for the suggestion. You perfectly mentioned that a good overlapping between CALIPSO and PRISMA for the same location and the same date of our case studies is almost impossible to get. However the emission rate for any of the studied facilities is clearly not steady. It changes on daily and even hourly basis. So we can't validate our results with such a comparison. Moreover CALIPSO/CALIOP has an horizontal resolution of 300 m (distance between acquisition along-track) and vertical resolution of 30 m. The probability to have a valid retrieval for both instruments is very low. Not to mention that the accuracy of the CALIOP retrievals in the lower part of the atmosphere depends on the above atmospheric transmission and on an adequate lidar ratio. Such an analysis is beyond the scope of our paper. However we might consider your suggestion for further investigation using hyperspectral imager and space lidar like EarthCare.**

**Minor comments and technical corrections**

The title mentions only the PRISMA hyperspectral imager whereas the authors use a synergy of PRISMA and S2/MSI multi-spectral observations, and it looks like the results shown in the article would not be achievable using only the PRISMA instrument. Wouldn't it be better to mention this synergy right in the title?

**Actually S2/MSI is used to constrain the surface reflectance and not to investigate plume properties. So to avoid any confusion, we prefer to keep the title as it is.**

Line 12 and elsewhere: normally, each number should be accompanied with its uncertainty. I hope that the uncertainties will be available after the first general comment above is addressed.

**Correct. The uncertainties on the mass flow rate are deduced from the uncertainties on the effective wind speed and the surface mass concentration as presented in section 6.1. The uncertainties are now given in the abstract, Table 3 and the section 6.1. Please note that the number given in the abstract and the Table 3 were not coherent. It's corrected now.**

Line 43: "though" -> "through"

Line 47: "impletation" -> "implementation"

Line 49: it would be better to use "smaller than" instead of "less than"

**Done. Thank you for your review.**

Line 105 and elsewhere: the spectral channels are mentioned, but no information on the kernels is given, whereas it would be interesting to see where does the information on aerosols come from.

**We have added the spectral range used on Line 202.**

Line 148 and the whole section: the regularized solution implies that there is a certain "safe" solution, around which we try to retrieve the result. How heavily is the solution regularized? What happens if the regularization coefficient is set to zero? Is the task so ill-conditioned that it won't converge at all? It would be interesting to compare the number of equations with number of unknowns

**The only regularization term (dumping factor) used here is in the Levenberg-Marquardt algorithm. The number of iterations is used to assess the convergence of the solution. The DOF are then checked to ensure that the solution is independent of the a priori.**

Line 232: it is not surprising that the farther the guess from the real solution is, the longer the iterative process is, but what is the link between the number of iterations and the footprint size, given that the pixels are treated individually?

**You are right. Each pixel is treated individually. Above 5 iterations, the size of the retrieved plume footprint becomes independent of the number of iterations. We use an empirical value of max 10 iterations to be sure to get all the pixels for which the algorithm converges.**

Line 238: does the solution depend on the initial AOT value?

**No. AOT is the component of the initial state vector having the weakest sensitivity (large associated uncertainty).**

Line 293: "to estimated" -> "to estimate"

**Done**